# Lanthanum Supplementation Alleviates Tomato Root Growth Suppression under Low Light Stress

**DOI:** 10.3390/plants12142663

**Published:** 2023-07-16

**Authors:** Syo Iguchi, Tatsuya Tokunaga, Eri Kamon, Yuto Takenaka, Shizuka Koshimizu, Masao Watanabe, Takeshi Ishimizu

**Affiliations:** 1College of Life Sciences, Ritsumeikan University, Kusatsu 525-8577, Japan; 2School of Agriculture, Meiji University, Kawasaki 214-8571, Japan; 3Graduate School of Life Sciences, Tohoku University, Sendai 980-8577, Japan

**Keywords:** lanthanum, low light stress, rare earth elements, root growth, tomato

## Abstract

Supplementation with rare earth elements (REEs) such as lanthanum and cerium has been shown to promote plant elongation and/or increase crop yields. On the other hand, there are reports that REE supplementation of plants has no such effect. The appropriate modes for REE utilization and the underlying mechanism are not fully understood. In this study, we investigated how REE supplementation of plants under low light stress affects plant growth and gene expression. Under low light stress conditions, tomato root elongation was observed to be reduced by about half. This suppression of root elongation was found to be considerably alleviated by 20 mM lanthanum ion supplementation. This effect was plant-species-dependent and nutrient-condition-dependent. Under low light stress, the expression of the genes for phytochrome-interacting factor, which induces auxin synthesis, and several auxin-synthesis-related proteins were markedly upregulated by lanthanum ion supplementation. Thus, we speculate that REE supplementation of plants results in auxin-induced cell elongation and alleviates growth suppression under stress conditions.

## 1. Introduction

Rare earth elements (REEs) belong to group 3 of the periodic table and are composed of 15 lanthanides, including lanthanum, as well as scandium and yttrium. Although REEs contain the word “rare”, the amount of lanthanum and cerium present in the Earth’s crust is not small and is comparable to the amount of copper and zinc. It is known that REEs have effects on living organisms. The addition of scandium or lanthanum to media for culture of *Streptomyces* and *Bacillus* species resulted in upregulated expression of secondary metabolite biosynthesis operons [1,2]. Methylotrophic bacteria, which were isolated from the surface of plants, require lanthanum for survival, and proteins that bind lanthanum ions (La^3+^), such as lanmodulin and malate dehydrogenase, have been found in these bacteria [3,4,5].

REEs also act on plants and have positive effects on plant growth and crop yield [6,7,8]. It was found more than 100 years ago that supplementation with low concentrations of lanthanum or cerium promoted cell division and increased petiole length in hyacinths [9]. It was also reported several decades ago that REE supplementation increased pea yield [10] and accelerated the growth of oat cotyledon sheaths [11]. Agricultural fertilizers containing REEs have been used in China since around the 1980s [8,12]. Appropriate concentrations of REEs (10^−6^–10^−5^ M) in fertilizers promote germination and root elongation [13,14]. However, some reports showed that even at such concentrations, REEs had no elongation-promoting effect [15]. It is not clear in what plants and under what conditions REEs act on plants to have a positive effect [6,7,8]. In addition, when REEs are applied to plants at concentrations around 10^−4^–10^−3^ M, elongation inhibition occurs [16,17,18].

The action of REE in plants at the molecular level are partially understood [6,7,8]. It has been shown that low pH or amino acids promote REE uptake into plants [19,20], REEs are incorporated into cell wall polysaccharides [21,22], and REEs increase peroxidase and nitrate reductase activity [23], inhibit redox homeostasis [24,25], increase the content of chlorophyll [24] or specialized metabolites [26,27], and induce cellular uptake by clathrin-dependent endocytosis [22,25,28,29]. However, the molecular mechanism for the action of REEs in plants is not fully understood.

To determine the conditions under which REEs act in plants to bring about positive effects and the underlying molecular mechanisms, we investigated the effects of REEs on plants under stress conditions. This knowledge will be useful in furthering our understanding of the molecular mechanisms of the REE actions in plants and in the development and use of fertilizers containing REEs. Low light is an environmental stress that affects plant growth [30,31]. In this study, we observed how REE supplementation of tomatoes grown under low light stress affected their root elongation and gene expression and attempted to elucidate some of the molecular mechanisms of REE action in plants.

## 2. Results

### 2.1. La^3+^ Supplementation Alleviates Suppression of Tomato Root Elongation under Low Light

Tomato seeds were sown on agar medium and at, 7 days after sowing, seedlings were transferred to lanthanum ion (La^3+^)-supplemented (0 to 200 μM) minimal-nutrient hydroponic medium (MGRL medium) [32]. At this time, root length was approximately 5 cm. The length of elongation was measured after transfer to hydroponic solution for 15 days at 25 °C, 16 h light, and 8 h dark conditions. At normal light intensity (photon flux density 132 ± 3 μmol m^−2^ s^−1^), during the 15 days of measurements, the mean increases in length of the roots and shoots were 8.7 cm and 17.8 cm, respectively (Figure 1A,C; red lines). At low light intensity (51 ± 4 μmol m^−2^ s^−1^), the mean increases in length of the roots and shoots were 3.9 cm (45% of normal light intensity increase) and 9.5 cm (53% of normal light intensity increase), respectively (Figure 1B,D; red lines). Thus, under low light intensity the elongation of tomato roots and shoots during the 15 days was reduced by half. Subsequent experiments were conducted under this low light stress condition.

At normal light intensity, 20 μM La^3+^ supplementation resulted in 110% elongation of roots (Figure 1A) and 108% elongation of shoots (Figure 1C) compared to no La^3+^ supplementation during the 15 days. La^3+^ supplementation at lower concentration (0.2 μM or 2 μM) had no effect on elongation, while 200 μM La^3+^ supplementation inhibited elongation; that is, elongation of roots was 62% (Figure 1A) and shoots was 74% (Figure 1C) compared to no La^3+^ supplementation. Thus, at normal light intensity, 20 μM La^3+^ supplementation promoted elongation by about 10%. Under low light stress with 20 μM La^3+^ supplementation, root elongation was 177% (Figure 1B) and shoot 129% (Figure 1D) compared to no La^3+^. That is, root elongation, which was suppressed to 45% under low light stress compared to that under normal light intensity, recovered to 82% with 20 μM La^3+^ supplementation (Figure 1A,B). Shoot elongation, which was suppressed to 53% under low light stress, recovered to 69% with 20 μM La^3+^ supplementation (Figure 1C,D). This indicates that La^3+^ supplementation alleviated the suppression of root and shoot growth caused by low light stress.

This alleviation was La^3+^-concentration-dependent. Under low light stress, with 2 μM La^3+^ supplementation, roots elongated 131% compared to no La^3+^ (Figure 1B). La^3+^ supplementation at a concentration of 0.2 μM had no effect on growth of roots or shoots, and 200 μM La^3+^ supplementation inhibited elongation of roots and shoots (72% and 78%, respectively, compared to no La^3+^) (Figure 1B,D). These results were obtained under the minimal nutritional conditions with MGRL medium [32]. In a replication of the measurements with higher nutrient conditions (1/2 Murashige and Skoog medium [33]) and under low light stress, the root elongation enhancement by 20 μM La^3+^ supplementation was 144% compared to no La^3+^ supplementation, which was lower than under the minimal-nutrient condition (177%; Figure 1B). This showed that the effect of REE supplementation was stronger in the lower nutrient conditions.

As the suppression of root elongation due to low light stress was alleviated by supplementation with 20 μM La^3+^ (Figure 1), we then investigated whether other REEs had the same effect by using 15 commercially available REE chlorides. All REEs tested alleviated elongation suppression by low light stress (Table 1). The optimal concentration of each REE was 0.2–20 μM (Table 1). Of these, La^3+^ was the most effective and Sc^3+^ was the next most effective. The REEs were classified into those that significantly reduced root elongation inhibition under low light stress (La^3+^, Sc^3+^, Ce^3+^, Pr^3+^, Nd^3+^, Dy^3+^, Er^3+^, and Tm^3+^) and those that reduce it to a lesser extent (Y^3+^, Sm^3+^, Eu^3+^, Gd^3+^, Tb^3+^, Yb^3+^, and Lu^3+^). A possible relationship between the stress suppression effect of REEs and their chemical properties was investigated [34] (Table 1) but no relationship could not be found. The most effective REE, that is La^3+^, was used in the following experiments.

### 2.2. Effect of La^3+^ Supplementation on Tomato Plant Growth under Low Light Stress

The cell length was measured at 3 cm from the root tip of tomato plants on days 0 and 15 after transfer to hydroponic media. Under normal light intensity, there was no significant difference in cell length between plants in media with and without 20 μM La^3+^ supplementation during the 15 days (mean cell length increased from 49 μm at day 0 to 67 μm at day 15) (Figure 2). In comparison, the mean cell length under low light stress increased from 46 μm to 52 μm without La^3+^ supplementation and from 48 μm to 62 μm with 20 μM La^3+^ supplementation. Thus, alleviation of root elongation suppression by La^3+^ supplementation under low light stress approximately corresponded to the change in root cell length.

Under low light stress, other characteristics were also altered by La^3+^ supplementation. Under normal light intensity with 0 and 20 μM La^3+^ supplementation, the numbers of lateral roots per plant were 18.4 and 23.3, respectively, and, under low light stress, 14.4 and 21.0, respectively (Figure 3A). This showed that La^3+^ supplementation also alleviated the stress-induced reduction in the number of lateral roots (Figure 3A). The concentrations of chlorophyll in extracts from tomato leaves without and with La^3+^ supplementation grown under normal light intensity were 22.4 μg/mL and 24.2 μg/mL and, under low light stress, 17.1 μg/mL and 23.6 μg/mL, respectively (Figure 3B). Thus, La^3+^ supplementation also improved chlorophyll levels that were reduced due to low light stress.

### 2.3. Analysis of Tomato Plants Grown in La^3+^-Supplemented Vermiculite

As we found that La^3+^ supplementation in a minimal-nutrient hydroponic medium alleviated the suppression of tomato root growth due to low light stress, we next examined the effect of La^3+^ supplementation on tomato growth in an oligotrophic vermiculite supplemented with MGRL solution. The mean increase in length of the shoots after 15 days at normal light intensity was 10.4 cm, and this did not change significantly with La^3+^ supplementation (Figure 4A). The mean increase in length the shoots grown under low light stress with 25 mg/kg La^3+^ supplementation was 7.8 cm; thus, the mean increase in length was 136% compared to no La^3+^ supplementation (Figure 4B). Thus, La^3+^ supplementation also alleviated low light stress in vermiculite medium. As in the hydroponic medium, this effect of La^3+^ supplementation in vermiculite was not observed when the eutrophic solution (1/2 Murashige and Skoog medium) was used instead of the minimal-nutrient solution. These results indicate that La^3+^ supplementation in vermiculite is also effective for plant growth improvement in low light stress and in relatively low-nutrient vermiculites.

### 2.4. Effect of La^3+^ Supplementation on Arabidopsis Root Elongation under Low Light Stress

The effect of La^3+^ supplementation to alleviate low light stress described above was observed in solanaceous tomato plants; we, therefore, investigated whether the same effects could be observed in a different plant family, cruciferous *Arabidopsis*. In *Arabidopsis*, normal light intensity was set at a photon flux density of 110 μmol m^−2^ s^−1^ and low light intensity at 20 μmol m^−2^ s^−1^. Root length elongation during 15 days was 12.3 cm under normal light intensity (Figure 5A) and 6.5 cm (53% of the elongation normal light intensity) under low light stress (Figure 5B). Under normal light intensity, no effect of La^3+^ supplementation on root elongation was observed (Figure 5A). A slight alleviation of low light stress was observed with 2 mM La^3+^ fertilization; root length was 7.3 cm with 2 mM La3+, 115% of that without La3+ (Figure 5B). Higher concentration (200 μM) La^3+^ supplementation inhibited root growth significantly. Thus, the suppression of elongation due to low light stress in cruciferous *Arabidopsis* was slightly reduced by La^3+^ supplementation but to a much lesser extent than that observed in solanaceous tomatoes. These results indicate that the alleviation of low light stress by La^3+^ supplementation varies among plant species.

### 2.5. Gene Expression of Tomato Roots Grown with La^3+^ Supplementation under Low Light Stress

As we found that La^3+^ supplementation alleviates the suppression of root elongation caused by low light stress, to explore the molecular mechanism of this effect, we analyzed gene expression in tomato roots grown with 0 and 20 μM La^3+^ supplementation using qRT-PCR. A total of 37 genes were analyzed (Appendix A), including antioxidant, endocytosis, photosystem, stress response, and plant-growth-related genes, whose expression was reported to be altered by La^3+^ supplementation in previous studies [24,25,29,35]. In Figure 6, values in column 1 are the fold changes in expression of 37 genes with 20 μM compared to 0 μM La^3+^ supplementation at normal light intensity, and values in column 2 are the corresponding changes in expression at low light stress. Column 3 is the values in column 2 divided by the values in column 1; thus, the values above 1 in column 3 indicate that the gene expression was upregulated with La^3+^ supplementation under low light stress. These were cell-growth-related extension, photosynthesis-related photosystem II reaction center protein, and auxin-related genes, including phytochrome-interacting factor 7, endocytosis-related clathrin light chain, and calcium-binding EF-hand protein (Figure 6, column 3). La^3+^ supplementation under low light stress induced changes in the expression levels of these genes, which was associated with alleviation of low light stress.

## 3. Discussion

We observed that 20 μM La^3+^ supplementation alleviated tomato root growth suppression caused by low light stress (Figure 1 and Figure 4). The ability of REEs to alleviate low light stress was particularly significant under low nutrient conditions.

The effect of La^3+^ supplementation on tomato root elongation was not so noticeable at normal light intensity, but its effect was greater under low light stress. The results presented here showing that the action of REEs on plants is more effective under low light intensity are consistent with the previous literature. At normal light intensity (100 to 200 μmol m^−2^ s^−1^), root elongation and chlorophyll content has been found to increase 1.1~1.2-fold with REE supplementation [13,29,36]. Moreover, at low light intensity (40 μmol m^−2^ s^−1^), chlorophyll content was about 1.5 times higher with REE supplementation [24]. Thus, there seems to be a general trend that the effects of REE supplementation are observed at lower light intensity.

The concentrations of REEs that were effective in alleviating low light-stress-induced elongation suppression ranged from 0.2 to 20 μM (Figure 1 and Table 1), which are similar concentrations to those reported in previous studies observing positive effects of REEs [6,7,8].

The effect of La^3+^ supplementation was observed in tomatoes of the Solanaceae family but not so much in *Arabidopsis* of the Cruciferae family (Figure 4). In the solanaceous sweet pepper, La^3+^ or Ce^3+^ supplementation was found to double the biomass [37,38]. On the other hand, the effect of La^3+^ supplementation was not so apparent in cruciferous *Arabidopsis* and mustard [39,40]. Thus, our results correspond to previous reports, and indicate that the responses to REE supplementation under stress conditions are dependent on the plant species.

The La^3+^ supplementation effects also depended on the nutrient conditions. The greatest effect of La^3+^ supplementation was observed when the plants were grown on minimal-nutrient media (Figure 1 and Figure 4). Under eutrophic conditions, the La^3+^ supplementation effect was low. When barley was grown under irrigated conditions, no La^3+^ supplementation effect was observed but, under drought conditions (water-deficit stress), La^3+^ supplementation increased water use efficiency and increased the number of tillers [41,42]. This means that La^3+^ supplementation alleviated the reduction in barley yield under drought-stress conditions. This effect of La^3+^ supplementation was observed only under low-nutrient conditions. This is the same as in the present study, where the effect of La^3+^ was observed under low-nutrient conditions and low light stress. Taken together, it appears that La^3+^ alleviates the stress response in plants that are under stress conditions in addition to low nutrients.

There have been many reports that REEs promote plant growth [13,14], as well as reports that they do not [15]. Considering the present results, it seems reasonable to assume that REEs have a positive effect on plants in response to stress only under certain conditions. The previously reported plant-growth-promoting effects of REEs may be simply observations that growth-suppressing stresses were alleviated by REEs. Fertilizers containing REEs are used in China, and these REEs may have a role in alleviating stresses such as poor nutrition, low light, and shortage of water.

One of the genes significantly upregulated by La^3+^ supplementation under low light stress was phytochrome-interacting factors (Figure 6). These are transcription factors that activate the auxin biosynthesis-related genes [43,44]. Consistent with this, auxin-related genes were also upregulated in the presence of La^3+^ under low light stress (Figure 6). PIF7 is known to interact with the light sensor phytochrome B and respond to shade (low light stress) [44]. This suggests a mechanism for the response to La^3+^ supplementation; that is, under low light stress, La^3+^ supplementation triggers a stress adaptation response, activates PIF-mediated auxin synthesis, and promotes auxin-mediated root elongation. It is known that responses to various stresses, such as salt, osmotic, and drought, lead to high expression of auxin-inducible transcription factors, increasing stress tolerance [45,46]. Thus, these results suggest that La^3+^ supplementation under low light stress leads to auxin biosynthesis, resulting in elongation promotion or the release of growth suppression.

REE supplementation activates clathrin-dependent endocytosis, which increases the Mg^2+^ and N content necessary for chlorophyll synthesis, resulting in an increased chlorophyll content [25,47]. In this study, the expression of genes for clathrin light chain, photosystem II reaction center protein, and chlorophyll *a-b* binding protein was particularly elevated with La^3+^ supplementation under low light stress (Figure 6). In addition, tryptophan, an auxin precursor, is actively synthesized using the increased N content, leading to increased auxin synthesis [48,49].

An La^3+^-binding protein, lanmodulin, has been identified in methylotrophic bacteria that were isolated from plant surfaces [5]. Lanmodulin has amino acid sequence homology with calmodulin. In this study, seeds were sterilized in each experiment, so it is unlikely that this bacterium and their lanmodulin contributed to the results of this study. Although lanmodulin has not been found in plants, it is known that REEs bind to two of the four metal-binding sites of calmodulin and that calmodulin responds to REE-induced endocytosis [50]. La^3+^ supplementation under low light stress increased the expression of some calcium-binding protein genes (Figure 6). It has been noted that calmodulin is associated with auxin-mediated cell expansion [51]. It is also possible that La^3+^ supplementation promotes elongation by such a mechanism.

From the results obtained to date, a putative molecular mechanism for the response to La^3+^ supplementation under low light stress may be proposed. That is, REE supplementation of plants under stress conditions may result in auxin-induced cell elongation or alleviation of growth suppression.

## 4. Materials and Methods

### 4.1. Plant Growing Conditions

The seeds of tomatoes (*Solanum lycopersicum* cv Momotaro) were purchased from Takii & Co., Ltd (Kyoto, Japan). The *Arabidopsis* ecotype used was Col-0 of *Arabidopsis thaliana*. The seeds were placed in 1.5 mL tubes, stirred with 1 mL of 70% ethanol for 1 min, and washed three times with 1 mL of distilled water. The seeds were then agitated with 1 mL of 10-fold diluted sodium hypochlorite for 20 min, washed three times with 1 mL of distilled water, and 2 μL of plant preservative mixture (Nacalai tesque, Kyoto, Japan) was added. The seeds were then placed at 4 °C for 1 day in the dark before sowing.

The seeds were sown and germinated on MGRL agar medium [32], which was composed of 1.5 mM MgSO_4_, 2.0 mM Ca(NO_3_)_2_, 3.0 mM KNO_3_, 67 μM Na_2_EDTA, 10.3 μM FeSO_4_, 6.5 μM MnSO_4_, 30 μM H_3_BO_3_, 1.0 μM ZnSO_4_, 24 nM (NH_4_)_6_Mo_7_O_24_, 130 nM CoCl_2_, 1.0 μM CuSO_4_, 2.3 mM MES, and 0.6% Gelrite (Fujifilm Wako Chemicals, Osaka, Japan). The pH was adjusted to 5.7 using KOH solution, and 50 mL of the medium was poured into a square petri dish (140 × 100 mm). One-week-old tomato seedlings were transferred to hydroponics or solid medium containing rare earth element chlorides, and their root and shoot elongations were measured for 15 days. For hydroponics, MGRL medium supplemented with each REE chloride (final concentration 0, 0.2, 2.0, 20, and 200 μM) was poured into a 1 L beaker and the pH was adjusted to 5.7 using KOH solution. One-week-old tomato seedings were put into 3 cm long straws (6 mm inner diameter), and these straws were inserted into a floating tube rack on hydroponics. The seedlings were fixed in the straws by inserting a piece of tissue paper. For solid medium, vermiculite containing LaCl_3_ at a concentration of 25 to 20,000 mg/kg was placed in germination pots (7.5 cm diameter) and MGRL solution was poured into the vermiculite. One-week-old tomato seedings were planted in this solid medium. Tomato seedlings were grown at normal light intensity (photon flux density 132 ± 3 μmol m^−2^ s^−1^) or low light stress (51 ± 4 μmol m^−2^ s^−1^) at 25 °C, 16 h light, and 8 h dark.

*Arabidopsis* seedlings were grown at normal light intensity (110 ± 2 μmol m^−2^ s^−1^) and low light stress (20 μmol m^−2^ s^−1^) at 22 °C, 16 h light, and 8 h dark for 15 days on MGRL agar medium.

### 4.2. Tomato Root and Shoot Lengths and Chlorophyll Content Measurements

The elongated lengths of tomato roots and shoots were measured for 15 days, starting on day 0, when the plants were transferred from the agar medium to the hydroponic or vermiculite medium. ImageJ was used to measure the length of tomato roots and shoots. Lengths of 30 samples per condition were measured and the mean and standard error were calculated. Chlorophyll was extracted by placing 0.05 g of tomato leaves cut with a cork borer (ø10 mm) in 1 mL of *N*,*N*-dimethylformamide overnight at 4 °C in the dark. The absorbance of the solution was measured at 646.8 and 663.8 nm, and the chlorophyll content was determined by the formula (Chl*a*+*b* = 17.67 × A_646.8_ + 7.12 × A_663.8_) [52]. The mean values and standard errors for 30 samples were calculated.

### 4.3. Observation of Tomato Root Cells

Tomato roots were fixed in FAA (formalin: glacial acetic acid: 95% ethanol: distilled water 10:5:50:35) for 1 h. A longitudinal section of the root was prepared by placing a root between slits in a 1 cm square piece of styrofoam and cutting with a razor blade. Sections were stored in TBS-T (25 mM Tris, 140 mM NaCl, 2.7 mM KCl, 1% Tween 20, pH 7.0) at 4 °C. Cell size was measured using an optical microscope BX53 (Olympus, Tokyo, Japan) and ImageJ. Fifty cells were measured, and the mean values and standard errors of their lengths were calculated.

### 4.4. Gene Expression Analysis Using qRT-PCR

RNA was extracted from tomato roots pulverized with a mortar and a pestle under liquid nitrogen using the RNeasy Mini Kit (Qiagen, Venlo, Netherland) [53]. qRT-PCR analysis was performed on RNA with RIN values greater than 7.0 measured using a Bioanalyzer 2100 (Agilent Technologies, Santa Clara, California). The cDNA was synthesized using PrimeScript Ⅱ 1st strand cDNA synthesis Kit (Takara Bio, Kusatsu, Japan). qRT-PCR was performed using StepOnePlus (Thermo Fischer Scientific, Waltham, Massachusetts). The mixtures were composed of 100 ng cDNA, 0.4 μM primer, 10 μL of TB Green Premix Ex Taq II, and 0.4 μL of ROX reference dye and were incubated at 95 °C for 30 s, (95 °C for 5 s, 60 °C for 30 s) for 40 cycles. Gene expression levels in tomato roots grown at normal light intensity without La^3+^ supplementation were set to 1. Relative expression levels were calculated using the ΔΔCt method [54] and compared by correcting the Ct values obtained from the target genes with those of the internal control GAPDH. Experiments were performed in triplicate, and means and standard deviations were calculated. The primers shown in Appendix A were designed to have amplified regions of 70–250 bp, primer lengths of 18–24, T_m_ values of 55 °C to 65 °C, and GC contents of 40–60%.

## Figures and Tables

**Figure 1 plants-12-02663-f001:**
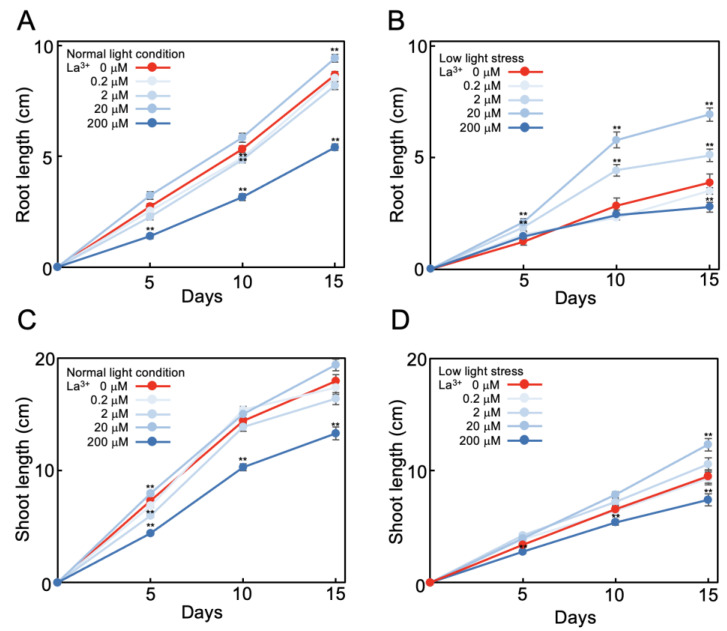
Effect of La^3+^ supplementation on tomato growth under low light stress. Elongated lengths of tomato root (**A**,**B**) and shoot (**C**,**D**) were measured during growth for 15 days in La^3+^-fertilized (0 to 200 μM) MGRL hydroponic solutions under normal light intensity (132 μmol m^−2^ s^−1^) (**A**,**C**) and low light stress (51 μmol m^−2^ s^−1^) (**B**,**D**). The growth conditions were 25 °C, 16 h light, and 8 h dark. Root and shoot lengths of 30 samples were measured and the means and standard errors were plotted. Statistical significances were determined by Student *t*-test compared with 0 μM (**, *p* < 0.01).

**Figure 2 plants-12-02663-f002:**
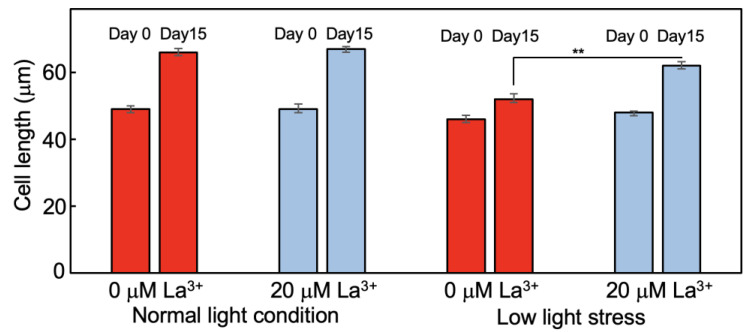
Cell lengths of tomato roots grown in La^3+^-supplemented medium under low light stress. Cell lengths observed in sections at 3 cm from the tip of tomato roots grown with and without La^3+^ supplementation under normal light intensity or low light stress. Roots were sampled at 0 and 15 days after transfer to the hydroponic medium. The lengths of 50 cells were measured and the mean and standard error were calculated. Statistical significance was determined by Student *t*-test (**, *p* < 0.01).

**Figure 3 plants-12-02663-f003:**
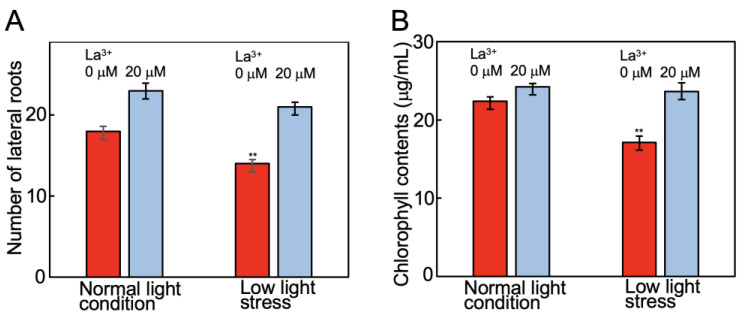
Effect of La^3+^ supplementation on lateral root number and chlorophyll content in leaves under low light stress. (**A**) Tomato lateral root number and (**B**) chlorophyll *a+b* content in tomato leaves grown with and without La^3+^ supplementation under normal light intensity or low light stress. In both cases, 30 samples were measured and the mean and standard errors were calculated. Statistical significances were determined by Student *t*-test compared with 0 μM La^3+^ under normal light condition (**, *p* < 0.01).

**Figure 4 plants-12-02663-f004:**
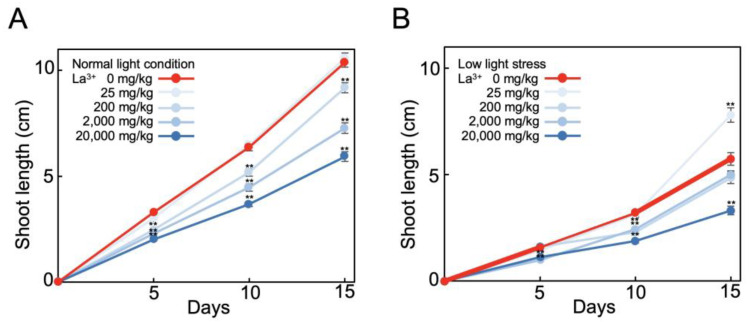
Growth of tomatoes in La^3+^-supplemented vermiculite. Elongated lengths of tomato shoots were measured for 15 days in La^3+^-supplemented vermiculite under normal light condition (**A**) or low light stress (**B**). The growth conditions were 25 °C, 16 h light, and 8 h dark. Shoot lengths of 30 samples were measured and the mean and standard errors were plotted. Statistical significances were determined by Student *t*-test compared with 0 mg/kg (**, *p* < 0.01).

**Figure 5 plants-12-02663-f005:**
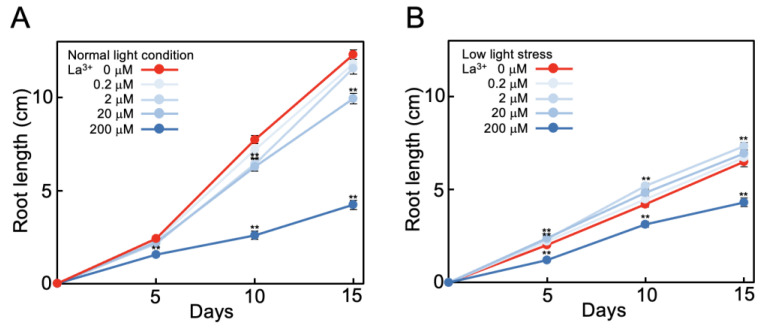
Effect of La^3+^ supplementation on *Arabidopsis* root elongation under low light stress. Elongated *Arabidopsis* root lengths were observed for 15 days in La^3+^-fertilized (0 to 200 μM) MGRL solid medium under normal light intensity (110 μmol m^−2^ s^−1^) (**A**) and low light stress (20 μmol m^−2^ s^−1^) (**B**). The growth conditions were 25 °C, 16 h light, and 8 h dark. Root lengths of 30 samples were measured and the means and standard errors were plotted. Statistical significances were determined by Student *t*-test compared with 0 μM (**, *p* < 0.01).

**Figure 6 plants-12-02663-f006:**
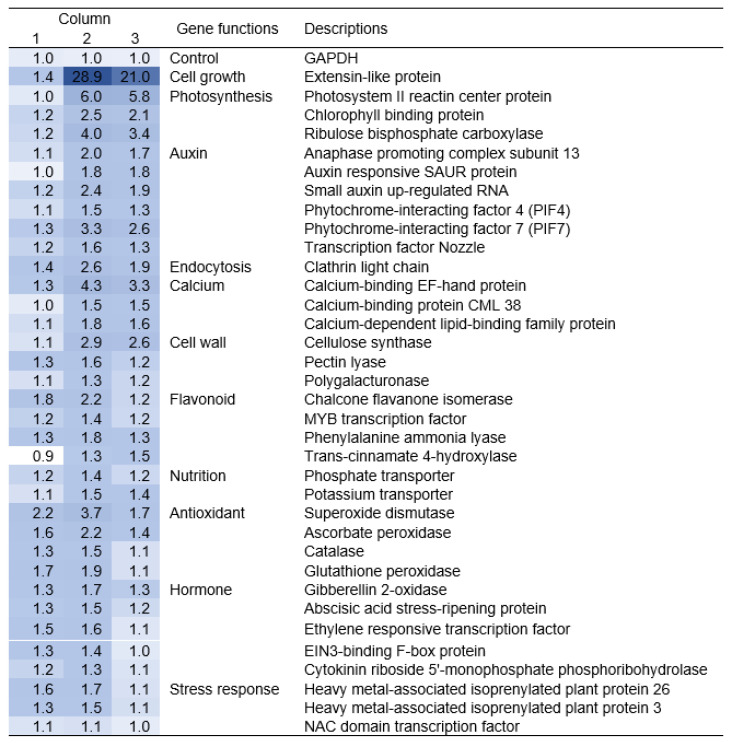
Gene expression by qRT-PCR in tomato roots grown with La^3+^ supplementation medium under low light stress. Gene expression levels in tomato roots grown without La^3+^ supplementation were set to 1. The fold changes in gene expression with La^3+^ fertilization were shown for roots under normal light intensity (column 1) and low light stress (column 2). Column 3 shows the fold change under low light stress (column 2) divided by the fold change under normal light (column 1). Dark blue indicates higher fold changes.

**Table 1 plants-12-02663-t001:** Effects of rare earth elements on elongation of tomato roots under low light stress and their chemical properties.

REE Ion	Optimum Concentration (μM)	Elongation with REE under Low Light Relative to Normal Light, % *	Elongation with REE Relative to No REE under Low Light, %	Atomic Weight	Ion Radius (Å)	Density (g/cm^3^)	Number of Unpaired 4f Electrons
None	–	45	100	–	–	–	–
Sc^3+^	20	72	161	45.0	0.75	2.99	0
Y^3+^	20	51	114	88.9	0.90	4.47	0
La^3+^	20	82	177	138.9	1.03	6.15	0
Ce^3+^	2	65	144	140.1	1.01	8.16	1
Pr^3+^	2	64	143	140.9	0.99	6.77	2
Nd^3+^	2	63	140	144.2	0.98	7.01	3
Pm^3+^	not available	–	–	145.0	0.97	7.26	4
Sm^3+^	2	52	115	150.4	0.96	7.52	5
Eu^3+^	0.2	48	107	151.9	0.95	5.24	6
Gd^3+^	0.2	50	112	157.3	0.94	7.90	7
Tb^3+^	20	50	112	158.9	0.92	8.23	6
Dy^3+^	0.2	62	138	162.5	0.91	8.55	5
Ho^3+^	not available	–	–	164.9	0.90	8.80	4
Er^3+^	20	65	145	167.3	0.89	9.07	3
Tm^3+^	0.2	60	133	168.9	0.88	9.32	2
Yb^3+^	2	52	116	173.0	0.87	6.97	1
Lu^3+^	2	52	115	175.0	0.86	9.84	0

* With no REE, root elongation was suppressed to 45% under low light stress. Values in this column above 45% indicate root elongation under low light stress was alleviated by REE fertilization.

## Data Availability

All data were provided within the article.

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
