# Peer review of "Lanthanum Supplementation Alleviates Tomato Root Growth Suppression under Low Light Stress"

_plants, 2023, doi:10.3390/plants12142663_

Round 1

Reviewer 1 Report

The functions of macro/micro-nutrients in plants have been intensively studied, however, the roles of rare earth elements in plants are still not fully understood. So, this paper consistent with the current research hotspot. But there are a few flaws in the paper which need to be revised.

1. The title named Lanthanum supplementation alleviates tomato root growth suppression under low light stress. However, in the introduction part, why choose low light as the stress to research?

2. The methods about Plant growing conditions is difficult to read. Please add the cultivate pictures to help the understanding.

3. The interesting finding of this study is that lanthanum significantly alleviated the stress of low light on tomato growth, and the auxin-related genes were also upregulated. Since plant photosynthesis is related to the light, the possible connections between photosynthesis and auxin should be more pronounced in the discussion part. 

Moderate editing of English language required. For example, 

"It is not fully understood under what conditions REEs have positive effects on plants and what molecular mechanism is involved" may be simpleliy as "The appropriate modes for REEs utilization and the underlying mechanism are not fully understood"   

Author Response

Response to Reviewer 1 Comments

The functions of macro/micro-nutrients in plants have been intensively studied, however, the roles of rare earth elements in plants are still not fully understood. So, this paper consistent with the current research hotspot. But there are a few flaws in the paper which need to be revised.

We are thankful to the reviewer for his/her critical comments. The changed parts in the manuscript are indicated in red letters.

  1. The title named “Lanthanum supplementation alleviates tomato root growth suppression under low light stress”. However, in the introduction part, why choose low light as the stress to research?

To investigate the effects of REEs on plants, this study analyzed the effects of REEs under stress conditions. Low light condition was chosen as the stress conditions. This is because low light is an environmental stress that affects plant growth and is suitable for this experiment, which was analyzed using plant growth as an indicator. This has already been described in lines 55-62.

  1. The methods about “Plant growing conditions” is difficult to read. Please add the cultivate pictures to help the understanding.

We would like to avoid adding the cultivate pictures. This is because they are not the main figures, yet they stand out. We have added a few words to the text to make it easier to understand "plant growing conditions." The changes made are in lines 68, 99, 171, 205, and 341. They are indicated in red. We believe this will help in understanding the experimental method.

  1. The interesting finding of this study is that lanthanum significantly alleviated the stress of low light on tomato growth, and the auxin-related genes were also upregulated. Since plant photosynthesis is related to the light, the possible connections between photosynthesis and auxin should be more pronounced in the discussion part.

This finding is very interesting. We have already discussed the relationship between light and auxin in lines 280-291. Thank you for your positive comments.

Comments on the Quality of English Language

Moderate editing of English language required. For example, 

"It is not fully understood under what conditions REEs have positive effects on plants and what molecular mechanism is involved" may be simpleliy as "The appropriate modes for REEs utilization and the underlying mechanism are not fully understood"   

Thank you for your comment. We rephrased this part as you suggested (Lines 14-15).

Reviewer 2 Report

Some additional, specific comments:

1. What is the main question addressed by the research?

the authors investigate how REE supplementation of plants under low light stress affects plant growth and gene expression

2. Do you consider the topic original or relevant in the field? Does it address a specific gap in the field?

The topic is relevant in this area for the whole world. This study significantly expands the possibilities of solving the scientific problem under study.

3. What does it add to the subject area compared with other published material?

Research demonstrates that REEs have a positive effect on plants in response to stress only under certain conditions (as opposed to unequivocal statements like "have a positive effect" or "haven't a positive effect" made in other studies)

4. What specific improvements should the authors consider regarding the methodology? What further controls should be considered?

I recommend avoiding phrases such as "we investigated...", instead it is better to write "the authors have investigated..." (or something similar)

5. Are the conclusions consistent with the evidence and arguments presented and do they address the main question posed?

Conclusions are logical and well founded

6. Please include any additional comments on the tables and figures.

The quality of tables and figures is appropriate

Author Response

Response to Reviewer 2 Comments

  1. What is the main question addressed by the research?

the authors investigate how REE supplementation of plants under low light stress affects plant growth and gene expression

That's right. We did these experiments and concluded that lanthanum supplementation alleviates tomato root growth suppression under low light stress.

  1. Do you consider the topic original or relevant in the field? Does it address a specific gap in the field?

The topic is relevant in this area for the whole world. This study significantly expands the possibilities of solving the scientific problem under study.

We agreed to this comment. We believe this study has expanded the effective use of REEs as fertilizers.

  1. What does it add to the subject area compared with other published material?

Research demonstrates that REEs have a positive effect on plants in response to stress only under certain conditions (as opposed to unequivocal statements like "have a positive effect" or "haven't a positive effect" made in other studies)

Yes. This is the major achievement of this study.

  1. What specific improvements should the authors consider regarding the methodology? What further controls should be considered?

I recommend avoiding phrases such as "we investigated...", instead it is better to write "the authors have investigated..." (or something similar)

The phrase "we investigated..." is frequently used in many papers published in the journal Plants. Therefore, we would like to leave this phrase unchanged.

  1. Are the conclusions consistent with the evidence and arguments presented and do they address the main question posed?

Conclusions are logical and well founded

  1. Please include any additional comments on the tables and figures.

The quality of tables and figures is appropriate

Thank you for your positive comments.

Round 2

Reviewer 1 Report

The responses are not quite comprehensive enough.

Point 1. The research points arise from the intersection between the protective mechanisms of Lanthanum and stress mechanisms of low light. Therefore, the mechanisms of low light stress on growth should be presented in detail, such as how it affecting nutrient uptake and photosynthetic. Then find the research intersections, and conduct the experiment purposely.

Point 2. If it is difficult to photo, the cultivation procedures can be painted.

(1) As the paper mentioned, different levels (0, 0.2, 2.0, 20, 200 uM) of REE were added in MGRL medium for hydroponics. Then LaCl3 at a concentration of 25 to 20,000 mg/kg (What the exactly treatments level are?) was placed in germination pots. What the relationship between these two procedures? Why not directly transfer the seedlings from clean environment to the La or REE exposure conditions? Let alone the authors once again mentioned “then transferred to hydroponic medium for 15 days.” in line 336.

(2) “One-week-old tomato seedings were put into 3 cm long straws (6 mm inner diameter), positioned with the straw around the hypocotyl, and inserted into a floating tube rack. They were fixed in the straws by inserting a piece of tissue paper.” These sentences are difficult to imagination.

No comments except that make the better presentation in 

Plant growing conditions

Author Response

Response to Reviewer 1 Comments

We are thankful again to the reviewer for his/her critical comments. The changed parts in the manuscript are indicated in red letters.

Point 1. The research points arise from the intersection between the protective mechanisms of Lanthanum and stress mechanisms of low light. Therefore, the mechanisms of low light stress on growth should be presented in detail, such as how it affecting nutrient uptake and photosynthetic. Then find the research intersections, and conduct the experiment purposely.

The point of this study is to find the conditions under which rare earth elements act on plants. We agree with your comment " The research points arise from the intersection between the protective mechanisms of Lanthanum and stress mechanisms of low light." We have mentioned it in the discussion (lines 280-291) as we did in our previous response. The reviewer gave us the following: "Therefore, the mechanisms of low light stress on growth should be presented in detail, such as how it affecting nutrient uptake and photosynthetic. Then find the research intersections, and conduct the experiment purposely". This point is a new research subject arising from the results of this study. This requirement is beyond the scope of this study. Therefore, we will not respond to this request. We would appreciate your understanding.

Point 2. If it is difficult to photo, the cultivation procedures can be painted.

(1) As the paper mentioned, different levels (0, 0.2, 2.0, 20, 200 uM) of REE were added in MGRL medium for hydroponics. Then LaCl3 at a concentration of 25 to 20,000 mg/kg (What the exactly treatments level are?) was placed in germination pots. What the relationship between these two procedures? Why not directly transfer the seedlings from clean environment to the La or REE exposure conditions? Let alone the authors once again mentioned “then transferred to hydroponic medium for 15 days.” in line 336.

(2) “One-week-old tomato seedings were put into 3 cm long straws (6 mm inner diameter), positioned with the straw around the hypocotyl, and inserted into a floating tube rack. They were fixed in the straws by inserting a piece of tissue paper.” These sentences are difficult to imagination.

Thank you for your comments. To make it easier for the reader to understand the experimental method, we have rewritten the paragraph in lines 321-340. We do not think it is necessary to provide illustrations because this is not a particular experimental procedure. We thank this reviewer for his/her suggestions because this manuscript has been improved.

The seeds were sown and germinated on MGRL agar medium [32], which was composed of 1.5 mM MgSO4, 2.0 mM Ca(NO3)2, 3.0 mM KNO3, 67 mM Na2EDTA, 10.3 mM FeSO4, 6.5 mM MnSO4, 30 mM H3BO3, 1.0 mM ZnSO4, 24 nM (NH4)6Mo7O24, 130 nM CoCl2, 1.0 mM CuSO4, 2.3 mM MES, and 0.6 % Gelrite (Fujifilm Wako Chemicals). The pH was adjusted to 5.7 using KOH solution, and 50 mL of the medium was poured into a square petri dish (140 × 100 mm). One-week-old tomato seedlings were transferred to hydroponics or a solid medium containing rare earth element chlorides, and their root and shoot elongations were measured for 15 days. For hydroponics, MGRL medium supplemented with each REE chloride (final concentration 0, 0.2, 2.0, 20, 200 mM) was poured into a 1 L beaker and the pH was adjusted to 5.7 using KOH solution. One-week-old tomato seedings were put into 3 cm long straws (6 mm inner diameter). These straws were inserted into a floating tube rack on hydroponics. The seedlings were fixed in the straws by inserting a piece of tissue paper. For solid medium, vermiculite containing LaCl3 at a concentration of 25 to 20,000 mg/kg was placed in germination pots (7.5 cm diameter), and MGRL solution was poured into the vermiculite. One-week-old tomato seedlings were planted in this solid medium. Tomato seedlings were grown at normal light intensity (photon flux density 132 ± 3 mmol m-2 s-1) or low light stress (51 ± 4 mmol m-2 s-1) at 25°C, 16 h light, 8 h dark.

Arabidopsis seedlings were grown at normal light intensity (110 ± 2 mmol m-2 s-1) and low light stress (20 mmol m-2 s-1) at 22°C, 16 h light and 8 h dark for 15 days on MGRL agar medium.

Round 3

Reviewer 1 Report

The authors has responded the comments accrodingly.  The exciting finding of this paper is the significant protecting role of REEs in plant growth. Hope they can continue this study to find more underlying mechnisms in more aspects, such as ionomics, phytohormone, ROS, etc, which are all related plant growth.

  •  

No comments.